# Learning Abstractions for Hierarchical Planning in Program-Synthesis Agents

## Abstract

Humans learn abstractions and use them to plan efficiently to quickly generalize across tasks—an ability that remains challenging for state-of-the-art large language model (LLM) agents and deep reinforcement learning (RL) systems. Inspired by the cognitive science of how people form abstractions and intuitive theories of their world knowledge, Theory-Based RL (TBRL) systems, such as TheoryCoder, exhibit strong generalization through effective use of abstractions. However, they heavily rely on human-provided abstractions and sidestep the abstraction-learning problem. We introduce TheoryCoder-2, a new TBRL agent that leverages LLMs' in-context learning ability to actively learn reusable abstractions rather than relying on hand-specified ones, by synthesizing abstractions from experience and integrating them into a hierarchical planning process. We conduct experiments on diverse environments, including BabyAI and VGDL games like Sokoban. We find that TheoryCoder-2 is significantly more sample-efficient than baseline LLM agents augmented with classical planning domain construction, reasoning-based planning, and prior program-synthesis agents such as WorldCoder. TheoryCoder-2 is able to solve complex tasks that the baselines fail, while only requiring minimal human prompts, unlike prior TBRL systems.

## 1 Introduction

A hallmark of human intelligence is the ability to plan hierarchically by combining abstract representations with a low-level world model (Koedinger & Anderson, 1990; Balaguer et al., 2016; Tomov et al., 2020; Correa et al., 2023). At an early age, infants understand abstract predicates like containment and support (Casasola & Cohen, 2002); these representations form the foundation of later skill development that enables humans to predict, manipulate, and plan in complex domains. For example, a representation of containment is essential for constructing an abstract plan to pour juice into a cup, which can then be combined with a low-level model of biomechanics and physics to construct a concrete plan grounded in the physical world.

Despite impressive progress, modern artificial intelligence (AI) systems still struggle to achieve comparable fluency with abstract planning. Taking inspiration from cognitive science (Gopnik & Meltzoff, 1997; Gerstenberg & Tenenbaum, 2017; Lake et al., 2017), recent work on "theory-based reinforcement learning" (TBRL) systems have sought to close this gap by endowing AI agents with human-like world models ("theories") that are object-oriented, relational, and causal (Tsividis et al., 2021; Tang et al., 2025). The most advanced system of this kind, TheoryCoder (Ahmed et al., 2025), learns a low-level world model, which is then combined with high-level abstractions to support hierarchical planning. The performance and sample efficiency of TheoryCoder on complex video games dramatically outstrips both deep reinforcement learning (RL) and large language model (LLM) agents. By harnessing LLMs to translate past experience into inferred world models, TheoryCoder also achieves high computational efficiency.

The main limitation of TheoryCoder is its reliance on hand-coded abstractions, which substantially limits its scope of application. Here we address this core limitation by implementing automated learning of abstract concepts. Our approach allows the agent not only to form and manipulate abstractions, but also to ground them effectively in new domains, enabling hierarchical planning for rapidly solving complex, novel tasks—emulating key algorithmic aspects of human learning and abstraction. The resulting method, TheoryCoder-2, is a TBRL agent capable of synthesizing high-level

abstractions in the form of "planning domain definition language" (PDDL; Ghallab et al. (1998)) operators, while requiring minimal human guidance in the form of initial prompts and examples.

We conduct experiments on several tasks based on video game description language (VGDL) games (Schaul, 2013), including Sokoban, as well as BabyAI (Chevalier-Boisvert et al., 2019) and Mini-Hack (Samvelyan et al., 2021) environments. We compare our method to several baselines based on LLMs augmented with classical planning domain construction (Liu et al., 2023a; Guan et al., 2023; Smirnov et al., 2024) and reasoning-based planning (Yao et al., 2023b; Wei et al., 2022; Yao et al., 2023a), as well as previously proposed program-synthesis agents such as WorldCoder (Tang et al., 2025).

We demonstrate that TheoryCoder-2 achieves substantial improvements in both sample-efficiency and generalization over the baselines: it can successfully solve complex versions of the tasks that the baselines fail. Overall, this represents a significant improvement in the applicability of TBRL, and an important step towards building AI systems that learn like humans.

## 2 BACKGROUND

### 2.1 THEORY-BASED REINFORCEMENT LEARNING

Theory-Based Reinforcement Learning (TBRL) is a paradigm in which an agent uses an explicit, program-like model of its environment and search algorithms to plan and solve problems. Unlike traditional model-based RL methods, which either encode dynamics of the environment as tabular transition models (Sutton, 1990; Kaelbling et al., 1996; 1998) or approximate them with deep neural networks (Schmidhuber, 1990; 2015; Pascanu et al., 2017; Weber et al., 2017; Ha & Schmidhuber, 2018; Hafner et al., 2020), TBRL systems represent the causal interactions between objects directly in the form of symbolic programs that describe how the environment works. TBRL is inspired by the cognitive theory in the sense that these programs are the *theories* corresponding to the abstract intuitive theories of the world that people learn and use for planning and problem solving. These systems are able to solve problems without having to rely on random exploration, since they try to uncover the causal relationships that have not been captured by their model. We provide a more concrete and formal description in the next Sec. 2.2.

The earliest concrete TBRL system, EMPA ("Exploring, Modeling, and Planning Agent"; Tsividis et al. (2021)), represented the environment using a domain-specific language, VGDL (Schaul, 2013), and employed Bayesian inference to generate them. EMPA was computationally slow due to the cost of inference; and VGDL itself was limiting, since it only allows expressing pairwise collision rules, making it difficult to scale beyond simple Atari-style domains.

More recently, TheoryCoder (Ahmed et al., 2025) advanced the TBRL paradigm by representing theories as Python programs—a general-purpose programming language, unlike VGDL—synthesized using large language models (LLMs; Brown et al. (2020)). LLMs enabled fast approximate inference in TheoryCoder, thereby also resolving the slowness issue of EMPA. TheoryCoder interacts with environments to collect inductive examples and use these to guide program synthesis. Importantly, it has introduced hierarchical (bi-level) planning by pairing low-level theories with high-level abstractions, which are themselves high-level theories. These abstractions were expressed in the Planning Domain Definition Language (PDDL; Ghallab et al. (1998); McDermott (2000)). These abstractions could be transferred across tasks, enabling rapid generalization and often enabling new levels of a game to be solved in just one or two interactions, resembling the efficiency of human learners. TheoryCoder has effectively achieved remarkable sample efficiency and generalization compared to other LLM agents on multiple 2D grid games, including Baba is You (Oy, 2019; Cloos et al., 2024). Further technical details of TheoryCoder are provided in the next Sec. 2.2.

Despite these promising results, the applicability of the current generation of TheoryCoder is still limited in that it i) is only applicable to environments that can be encoded through object-oriented coordinate-based representations, and ii) heavily relies on hand-engineered abstractions, limiting scalability.

The main technical contribution of our method (Sec. 3) is to address the latter, by enabling the TBRL system to learn abstractions automatically in a few-shot manner. This capability allows TBRL agents not only to reuse abstractions across different games but also to expand its repertoire of theories

in entirely new domains—capturing the human ability of gradually growing a library of reusable structured concepts—without manual intervention. We show our approach substantially improves computational efficiency (in terms of tokens used) over the baselines of existing LLM-based agents, and accelerates learning speed, making it a stronger step toward scalable, human-like abstraction learning and problem solving through TBRL.

## 2.2 THEORYCODER

Here we describe the mathematical details of TheoryCoder (Ahmed et al., 2025), which we directly build on. The problem is formulated as follows. The environment is modeled by a transition function $T : \mathcal{S} \times \mathcal{A} \to \mathcal{S}$ over state space $\mathcal{S}$ and action space $\mathcal{A}$. Let $K$ denote a positive integer. The agent's objective is to find a plan $\pi = (a_1, \ldots, a_N)$ with $a_i \in \mathcal{A}$ for all $i$ from 1 to $N$, that minimizes cumulative cost $\sum_{n=1}^{N} c(s_n, a_n)$, where $c(s, a) = K$ for non-goal states and $c(s^*, a) = 0$ at the goal state $s^*$. Thus, an optimal plan corresponds to the shortest action sequence from the start state to the goal.

**System overview.** TheoryCoder is an agent consisting of five components: an LLM, two planners (for high-level and low-level planning, respectively) and a set of PDDL program files representing a library of abstract states and actions (which are used by the high-level planner), as well as a Python program file representing the world model which approximates the transition function of the environment (which is used by the low-level planner). The LLM and planner components are pre-specified when defining the system, and remain fixed. Essentially, learning in TheoryCoder consists in synthesizing these program files (using the LLM) while interacting with the environment; planning consists in executing the classic PDDL planning system and search algorithms using these program files. The complementary roles of these program files are further described below.

**High-level abstractions.** The PDDL program files in TheoryCoder contain the agent's current library of high-level abstract domain theories. To be more specific about their structure, these PDDL program files consist of a "domain" file and a "problem" file. The domain file specifies abstract actions (called "operators", e.g., "open door") and their preconditions/effects, as well as abstract states (e.g., "door unlocked") which are represented by Boolean predicates that capture task-relevant features. The problem file specifies the initial state and goal conditions for a particular task. Together, these files are taken as input by a classical planner (we use Fast Downward (Helmert, 2006)), which outputs a plan (a sequence of operators, i.e., high-level actions) that achieves the goal from the initial state, if one exists. In the original TheoryCoder, these PDDL files are assumed to be given by the human engineer. Similarly, EMPA depended on a set of VGDL abstractions that it was provided. This dependency on unlearned abstractions is the limitation we address here.

**Low-level dynamics world model.** TheoryCoder maintains an additional Python program $\hat{T}$ generated by prompting the LLM, representing the environment's transition function (world model), which fully predicts the effects of low-level actions in the raw state space. Python programs are generated using zero-shot prompting, where the instructions are to revise the current code to correct a set of prediction errors, which are provided in the prompt. Proposed revisions are evaluated against ground truth from the replay buffer, and if any errors remain, the LLM is re-prompted until they are fixed (up to a fixed budget). The agent is always prompted to revise its code whenever prediction errors occur. Observations are generated whenever the agent takes actions, and these are stored in a replay buffer. When an agent first begins a new task, it is allowed a small amount of random exploration in the low-level action space (e.g. "up", "down", "left", "right"), which generates an initial set of observations. The low-level world model is initialized as an empty function, which predicts no changes in state as a result of any action.

**Bi-level planning.** Once the PDDL domain and problem files are generated, the high-level planner (Fast Downward; Helmert, 2006) generates a symbolic plan in terms of abstract operators. The low-level planner (in this case breadth-first search) uses the learned transition function $\hat{T}$ to map each operator to a sequence of primitive actions. A bridging "checker" function ensures consistency by verifying that the low-level states indeed satisfy the intended high-level predicate effects. This hierarchical framing mirrors how humans plan: abstract operators (e.g., "open door") and predicates (e.g., "door unlocked") guide planning at the symbolic level, while grounding them requires concrete motor actions (e.g., "up", "left", etc.).

# 3 METHOD: THEORYCODER-2

We extend TheoryCoder by enabling it to autonomously learn abstractions (i.e., synthesize the PDDL files) and grow a library of abstract concepts/skills through a sequence of episodes interacting with various environments. We refer to this improved TBRL system as TheoryCoder-2. Here we describe the details of the abstraction learning process of TheoryCoder-2 via LLM in-context learning (Sec. 3.1) and the overall idea of gradually growing the library of abstractions through a curriculum (Sec. 3.2). An algorithmic description is presented in Algorithm 1.

---

**Algorithm 1** TheoryCoder-2

---

**Require:** LLM, initial state $s_0$, action space $\mathcal{A}$, few-shot examples $FS$
**Ensure:** Low-level plan $\pi = \langle a_1, a_2, \ldots, a_N \rangle$
1: $D, \mathcal{P} \leftarrow \text{LLM}(s_0, FS)$  ▷ Generate PDDL domain $D$ and problem $\mathcal{P}$ via few-shot prompting
2: $\mathcal{R}_p \leftarrow \emptyset, \mathcal{R}_a \leftarrow \emptyset$  ▷ Initialize replay buffers
3: $\mathcal{R}_{\text{random}} \leftarrow$ Generate transitions with random actions
4: $\hat{T} \leftarrow \text{LLM}(s_0, \mathcal{A}, \mathcal{R}_{\text{random}}, D, \mathcal{P})$  ▷ Initialize transition model
5: $\mathcal{G} \leftarrow \text{LLM}(s_0, FS, D, \mathcal{P})$  ▷ Generate Python predicates that define the EFF($\underline{\omega_k}$) of operators
6: $\Pi_H \leftarrow \text{Fast-Downward}(D, \mathcal{P})$  ▷ Obtain high-level plan
7: **for** each grounded operator $\underline{\omega_k} \in \Pi_H$ **do**  ▷ Low-level planning loop
8:    $\pi_k \leftarrow \text{BFS}(s_0, \hat{T}, \underline{\omega_k})$  ▷ Find low-level actions satisfying EFF($\underline{\omega_k}$)
9:    Store $(s, a, s', \underline{\omega_k})$ for all transitions in $\pi_k$ in $\mathcal{R}_p$
10:    $\pi \leftarrow \pi \cup \pi_k$  ▷ Append low-level subplan
11:    **for** each $a \in \pi_k$ **do**  ▷ Execute actions in environment
12:       Store $(s, a, s', \underline{\omega_k})$ in $\mathcal{R}_a$
13:    **end for**
14: **end for**
15: **if** EFF($\omega_{K-1}$) holds in $s'$ after executing $\pi$ **then**  ▷ Check if final goal conditions are met
16:    **return** $\pi$
17: **else**
18:    **if** $\mathcal{R}_p \neq \mathcal{R}_a$ **then**  ▷ Check for mismatches
19:       $\hat{T} \leftarrow \text{LLM}(\hat{T}, \mathcal{R}_p, \mathcal{R}_a)$  ▷ Refine transition model
20:    **else**
21:       $\Pi_E \leftarrow \text{Exploration}(s_0, D)$  ▷ Generate new high-level plan, see Section 3.2
22:       **goto** Line 8, execute $\Pi_E$ once, then re-execute $\Pi_H$  ▷ $\Pi_E$ will be used to refine $\hat{T}$
23:    **end if**
24: **end if**

---

## 3.1 LEARNING ABSTRACTIONS

Unlike the original TheoryCoder, which relied on hand-engineered PDDL files defining abstract operators and predicates, TheoryCoder-2 leverages LLMs' in-context learning ability to synthesize such files on its own. In this process, the only system input we need to hand-engineer is the *initial prompt*—a natural language description of the final goal of the environment and very simple examples illustrating what it means to learn abstract operators (e.g., *eat* with the precondition *not eaten* and the effect *eaten*) based on a toy problem; the corresponding example can be found in Box 4 and the complete initial prompt in Box 2 in the Appendix.

These examples are designed to be minimal; they are unrelated to the actual environment the agent interacts with, serving only as templates for how abstractions can be represented. They are crucial for guiding the LLM to synthesize abstractions at the appropriate level—neither too granular nor too coarse (a challenge for current LLMs when given full autonomy). In practice, we found that no more than one example is necessary for the agent to then form abstractions in new environments.

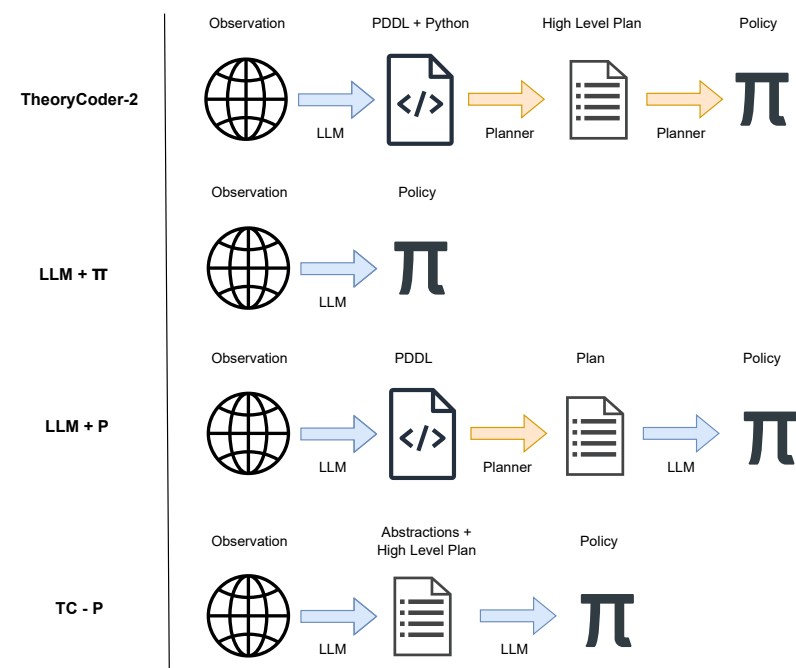

Figure 1: Comparison of agent–environment interaction between methods. WorldCoder goes through the same process as LLM + P except it synthesizes a Python file instead of PDDL files.

## 3.2 REUSING AND GROWING THE LIBRARY OF ABSTRACTIONS

Once the mechanism for learning abstractions is in place (Sec. 3.1), another crucial ability of abstraction learners is to *reuse* such abstractions and, if necessary, to continually generate and add new ones to the existing library of abstractions, while interacting with new environments.

TheoryCoder-2 gradually learns and grows a library of abstract concepts through a sequence of "episodes" interacting with different environments. Each episode can contain one or more environments that are grouped together by similarity. We assume access to a curriculum in which the agent begins with the easiest environment and progresses to increasingly harder ones. Nevertheless, our ablation shows that while such a curriculum improves sample efficiency, it was not essential for the success of the abstraction-learning process itself in the domains studied here.

The agent starts with the simplest game and generates a PDDL domain and problem file for it. Given that the curriculum is ordered and similar games are grouped together, once the agent has successfully generated useful abstractions in the current environment, it may reuse them to quickly solve the next few games. For example, if the main skill required to solve the first environment is navigating to a goal, the agent may synthesize the PDDL operator move_to and use it to generate high-level plans for moving to the goal. If the environments in the next episode require the same skill, it can reuse it, while learning new skills needed for the new environments, and so forth.

Within each episode, once the agent has learned the operators, it then learns a Python world model exactly as in the original TheoryCoder (Sec. 2.2). In addition, it learns how to connect the Python transition function and the PDDL abstractions by writing Python functions to map the PDDL predicates onto the low-level (raw) state. That is, the agent learns the predicate classifiers using Python. These classifiers are crucial because they are used to check whether an observation satisfies a given predicate. If the agent encounters a similar environment where an abstraction can be reused then only the low-level transition dynamics model is continually refined from experience. As we will show empirically for some of the VGDL games, TheoryCoder-2 is also able to reuse the dynamics model. Once all the main files are generated, the high-level planner will return a high-level plan and for each high-level plan the low-level planner will find the action sequence to be directly executed in the environment. We provide all of our prompts in the appendix.

## 4 EXPERIMENTS

Our experiments aim to answer the following set of questions: Can TheoryCoder-2 successfully learn abstract states and actions? Are learned abstractions reusable across different environments? Does reuse improve sample efficiency on new problems? How well does the resulting system perform on challenging tasks that are nontrivial for existing LLM agents?

To answer these questions, we evaluate various properties of TheoryCoder-2 and other agents in two experimental settings: VGDL-based games: Labyrinth, Maze, and Sokoban (Sec. 4.1) and BabyAI environments (Sec. 4.2)—each designed to evaluate a key capability of TheoryCoder-2 in isolation.

**Evaluation metrics.** We use the following metrics to evaluate agents: *token cost* (the number of tokens consumed by each agent measures sample efficiency), *compute time* (Wall-clock compute time measures the practical runtime of each agent), and *solution rate* (the proportion of tasks (game levels) successfully solved on the first attempt measures agent performance).

We compare TheoryCoder-2 against the following baselines, including variation of TheoryCoder-2 in which we ablate certain components. A visual comparison between these systems can be found in Figure 1. All of these agents use LLMs in some capacity (either the non-reasoning model 4o or the reasoning model o4-mini).

**LLM + $\pi$.** A reasoning-only model that generates plans directly in terms of primitive actions, without explicit abstractions or an executable world model. Here, we test o4-mini (OpenAI, 2025), with high, medium, and low reasoning effort (when not indicated, we use the 'high' variant). Additionally, we test GPT-4o, a non-reasoning model.

**LLM + P** (Liu et al., 2023a). Uses an LLM to generate PDDL domain and problem files for each task given the current observation and a few-shot prompt. A planner produces a plan, which an LLM then converts into a sequence of actions that are executable in the environment. We use o4-mini with high reasoning effort as the PDDL synthesizer model, since lower-effort modes struggled even on the earlier levels—likely because LLMs are less reliable at producing PDDL than Python code.

**WorldCoder** (Tang et al., 2025).The agent synthesizes a Python program representing the transition function (therefore, this could have been denoted as "LLM + Py" in our terminology). This program is used by a low-level planner to generate actions. Just as for TheoryCoder-2, we use GPT-4o as the synthesizer and BFS as the planner. WorldCoder differs from TheoryCoder-2 in that planning and world modeling is not done hierarchically. WorldCoder is more similar to LLM + P in that aspect, as LLM + P is also modeling the world at a low-level only.

**TheoryCoder-2.** Our full system, which synthesizes PDDL operators using GPT-4o for high-level planning, along with Python versions of the predicates and a Python transition function for low-level dynamics, enabling grounded abstraction learning and reuse across environments. TheoryCoder-2 is different from the other agents since it models the world hierarchically by synthesizing high level abstractions in PDDL and a low-level Python transition model. It uses the PDDL operators for high-level planning and the low-level model for low-level planning.

**Oracle**. Uses hand-engineered abstractions, serving as a reference to compare the quality of abstractions learned by TheoryCoder-2.

We additionally evaluate two ablated variations of TheoryCoder-2:

**TC - P.** Removes the executable abstractions and Python world model. The LLM directly outputs abstractions and high-level plans, and then is prompted to convert its high level plan into actions.

**TC - C.** Removes curriculum learning. It starts each episode with blank abstractions and transition function. It has to synthesize all the abstractions and a transition function for the current level.

### 4.1 EVALUATING ABSTRACTION LEARNING AND REUSE IN SIMPLE ENVIRONMENTS

The goal of this experiment is to evaluate the feasibility of abstraction learning and its reusability in our agent. Here we evaluate the agents using **Labyrinth**, **Maze**, and **Sokoban**. The first segment of Figure 2 illustrates this setting. These tasks are navigation-style VGDL games that primarily involve learning and reusing abstractions to "move to" a certain position.

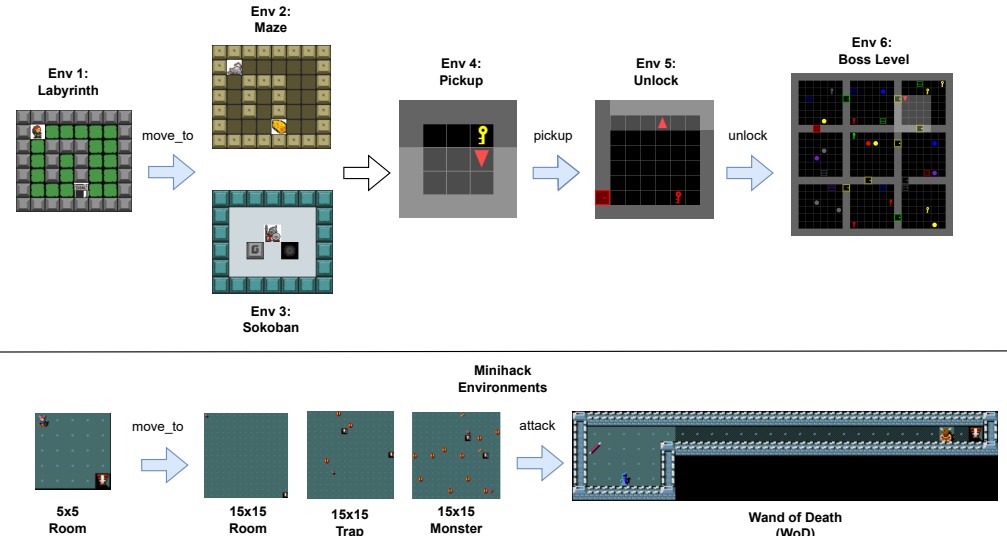

Figure 2: An illustration of the curriculum used in our experiments. A curriculum is a sequence of episodes in which each episode contains one or more environments/games. The sequence of the first episode (Labyrinth) and the second one (Maze, and Sokoban) is studied in Experiment 1 (Sec. 4.1), while the entire sequence is used in Experiment 2 (Sec. 4.2). In experiment 3 (Minihack), we use a separate curriculum. The blue arrows indicate the abstractions that TheoryCoder-2 learned.

Results in the top part of Table 1 show the token cost and whether the agents successfully solved each of the problems. First, we observe that TheoryCoder-2 is able to learn the key abstraction move_to and solve the task. (Note the LLM named this operator moveontop and the corresponding code for the abstraction is shown in Box 1 below.) Second, we observe that TheoryCoder-2 was able to reuse this operator in two new environments, Maze and Sokoban. In terms of efficiency, the simple LLM + $\pi$ baseline is the most efficient agent on these simple environments, while the second best is TheoryCoder-2 outperforming the two advanced LLM agents, LLM + P and WorldCoder. Finally, we note that all systems were able to solve these simple problems.

---

**Box 1: Example of a learned abstraction (implementing an operator to move to a certain object)**

```
(:action moveontop
    :parameters (?obj1 – object ?obj2 – object)
    :precondition (not (ontop ?obj1 ?obj2))
    :effect (ontop ?obj1 ?obj2)
)
```

---

## 4.2 TRANSFERRING LEARNED ABSTRACTIONS TO HARDER PROBLEMS

The purpose of this experiment is to test whether TheoryCoder-2 can gradually learn new abstractions and reuse them in new environments, and whether that yields sample efficiency.

We first evaluate our agent on the curriculum of Sec. 4.1; we add three **BabyAI** levels in sequence: *Pickup* (single key), *Unlock* (key + door), and the *Boss* level. The two first environments are named after the abstract skills needed to solve the corresponding task, and the last one is a multi-room task requiring both picking-up and unlocking skills. For the Boss level, we generate three instantiations with different layouts (based on three different seeds) in order to increase the diversity of final combined-skill environments. Figure 2 provides an illustration of this curriculum.

Table 1 (middle) shows the results. While all the agents solve the simple *Pickup* and *Unlock* environments, many of them fail in the complex Boss level: more specifically, all agents failed in "Combined Skills 2", while only TheoryCoder-2 succeeded at both "Combined Skills 1" and "3".

Table 1: Token cost across models (lower is better). Cells are highlighted in blue if the corresponding agent **failed** to solve the task.

| Task (Game) | TheoryCoder-2 | | | Baselines | | |
|---|---|---|---|---|---|---|
| | Full | TC - P | TC - C | LLM + $\pi$ | LLM + P | WorldCoder |
| Labyrinth | 21,378 | 24,510 | 21,378 | 5,173 | 28,931 | 56,360 |
| Maze | 19,737 | 23,186 | 21,236 | 3,518 | 24,396 | 56,085 |
| Sokoban | 7,171 | 10,373 | 8,441 | 2,608 | 25,919 | 19,684 |
| **BabyAI** | | | | | | |
| BabyAI (Pickup) | 8,588 | 6,660 | 8,588 | 2,405 | 20,589 | 18,013 |
| BabyAI (Unlock) | 33,116 | 41,734 | 33,116 | 5,705 | 50,071 | 97,938 |
| BabyAI (Combine Skills 1) | 1,961 | 54,277 | 44,725 | 40,960 | 41,515 | 119,330 |
| BabyAI (Combined Skills 2) | 2,528 | 53,376 | 45,175 | 49,973 | 59,003 | 120,200 |
| BabyAI (Combined Skills 3) | 2,454 | 53,064 | 45,017 | 29,791 | 55,078 | 120,375 |
| **Minihack** | | | | | | |
| Minihack-5x5 | 5,163 | 7,671 | 5,163 | 1,115 | 12,595 | 8,144 |
| Minihack-15x15 | 0 | 9,815 | 4,837 | 1,402 | 12,124 | 0 |
| MiniHack-Traps | 0 | 14,326 | 5,007 | 9,110 | 29,712 | 0 |
| MiniHack-Monster | 0 | 21,189 | 6,125 | 1,290 | 30,940 | 0 |
| Minihack-WoD | 19,433 | 21,932 | 19,433 | 4,176 | 52,434 | 62,165 |
| **Total for All Tasks** | **121,529** | 267,180 | 268,241 | 157,206 | 443,307 | 437,719 |

TheoryCoder-2 successfully learns abstractions (`Pickup` and `Unlock`) from the two first levels and then composes them to solve the more complex Boss level tasks that require using both of them.

In terms of sample efficiency, TheoryCoder-2 allocates high computation for the first and the second levels to learn the abstractions; but its token consumption dramatically drops on the Boss level (from about 8500 and 33000 to around 2000), as it no longer needs to learn any new abstractions if it reuses those learned in the previous level. Furthermore, it shows that TheoryCoder-2 is able to compose these primitive abstractions to solve different variations of the Boss levels that have different win conditions.

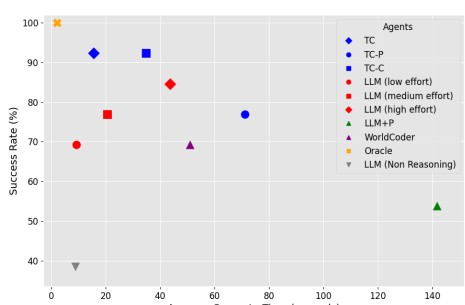

Figure 3: Success rate as a function of compute time, averaged across all games. The TC Family represents TheoryCoder-2 and its variants. TheoryCoder and its ablations are able to solve more tasks with less compute time than the reasoning models that use high reasoning effort. LLM + $\pi$ is shown with three different reasoning efforts.

Finally, we evaluate our abstraction learning method in five Minihack environments. The bottom part of Table 1 illustrates the effectiveness of abstraction transfer. TheoryCoder-2 learns an abstract skill of `move_to` in `Minihack-5x5` and then reuses it to solve the next three environments. As a result, there is 0 token consumption for `Minihack-15x15`, `Minihack-Traps`, and `Minihack-Monster`. WorldCoder also exhibits this trend, but failed at the hardest task, `Minihack-WoD`. In the `Minihack-WoD` environment, TheoryCoder-2 learns the `attack` abstraction, which allows it to shoot a wand to kill an enemy. This abstraction demonstrates how compact operators that encompass multiple concepts can be learned: picking up the wand, aiming it at the enemy and shooting it. The Minihack experiments demonstrate the effectiveness of zero-shot transfer of the abstractions as well as the ability to learn single compact abstractions that encompass different actions.

Notably, even when the curriculum or planners is removed, TheoryCoder-2 is capable of solving the task; that is, both components significantly contribute to improving the sample efficiency but not to the performance. In Table 1, we see that even when curriculum learning is removed and abstraction files are initialized with blank files, TheoryCoder-2 remains more compute efficient than the other world modeling approaches: WorldCoder and LLM + P. By contrast, WorldCoder is very costly, consuming more tokens than the raw LLM approach. On the more difficult environments, we also

observed that the PDDL programs generated by LLM + P frequently contained errors, leading to invalid or unsolvable plans.

We further compare the runtime of these agents. Fig. 3 shows the success rate averaged over all games in the curriculum and the average compute time. We observe that the fastest agents are the low-effort LLM + $\pi$ baseline and the full TheoryCoder-2 with curriculum and planner. Notably, even when curriculum learning is removed and TheoryCoder-2 is initialized with blank files, it remains faster at synthesizing abstractions and solving levels than the o4-mini variants.

## 5 DISCUSSION

**Results summary.** The results highlight several key advantages of TheoryCoder-2 over the baselines. As shown in Fig. 3, TheoryCoder-2 and its ablations achieve the highest success rate. This stems from the use of grounded abstractions, which reduce the likelihood of planning errors compared to reasoning-only LLMs. While LLM + $\pi$ with high reasoning effort sometimes achieve comparable solution rates, it does so at a higher token cost and much more compute time cost, making it impractical for scalable or real-time use. On the BabyAI Boss level, o4-mini with high reasoning effort often required around 3 minutes to return an answer. In contrast, TheoryCoder-2 invests compute in simpler levels to learn reusable components of hierarchical world models, which then accelerate learning and planning in harder environments. Moreover, by generating symbolic abstractions, our agents can run highly efficient classical planning algorithms. The result is efficiency gains in both token cost and runtime: symbolic planners typically terminate within a second in the domains we explored here.

With the TC-P ablation, which both generated abstractions and planned in natural language instead of code, we observed qualitative differences compared to the full model. For example, o4-mini with high reasoning effort produced abstractions at a different level of granularity than our system, often interleaving high-level operators with unnecessary low-level details. Thus, more strictly enforcing clean separation of function between levels of the world model via code specifications may have more general benefits for producing systematic thinking in LLM agents.

Our results suggest that the mixing in of low-level detail may be why TC-P takes longer to map out plans, whereas TheoryCoder-2 invests just enough compute up front to synthesize an appropriate world model. Once this model captures the "right" abstractions, it can serve as an adaptive compute resource, allowing the agent to flexibly balance fast, reactive reasoning with slower, more deliberate planning depending on the situation.

Overall, we found TheoryCoder-2's learned abstractions to be of similar quality to the Oracle, the hand-designed abstractions. While there is no systematic way to quantitatively evaluate the quality of the abstractions learned (e.g., one could measure the string length of programs and ask how compressed a particular abstraction is), qualitatively, we find that they yield similar performance to the Oracle in Fig. 3 and have similar preconditions and effects.

**Limitations and future directions.** Despite the significant advances, TheoryCoder-2 and TBRL architectures still have limitations, which we plan to address in future work. First, our approach assumes access to an object-oriented, text-based state representation. While vision-language models have shown mixed results for planning, they may serve as perception modules for extracting such representations in simple environments; scaling to more complex settings will require robust methods for object discovery, tracking, and attribute inference. Second, extending beyond discrete domains to continuous ones introduces new challenges such as modeling physics (e.g., predicting velocities and contacts). Third, we noticed issues related to brittleness when learning the predicate classifiers, which were critical for linking high and low levels of representation in the planner, or edge cases not covered by the learned abstractions. In particular, we observed these problems in the boss levels of BabyAI, which included multiple doors, where TheoryCoder-2 occasionally failed (see Table 1, "Combine Skills 2"). BabyAI's "Combined Skills 2" is challenging because it requires traversing a multi-room layout to reach a room on the opposite side. Along the way, the agent must correctly infer that boxes and balls should be picked up and moved aside to clear each doorway. This can be evaluated further by giving our agent more iterations to revise its world model.

Finally, we note an important direction of ongoing work. The experiments we presented here were limited in that agents generated abstractions for each domain once, at the beginning of their interactions in a particular domain, and did not revise them in light of new observations. We are currently

developing methods for revising abstractions through trial-and-error. One technique we are developing is to use previously learned operators, such as move_to, as a bootstrapping method to generate informative observations. Exploration patterns produced using high-level abstractions are likely to be much more informative than completely random exploration, even if those abstractions are not adequate for solving the domain. We predict that augmenting TheoryCoder-2 in this way will further enhance agents' ability to solve complex tasks.

## 6 Related Work

**LLMs for Planning and Synthesizing Policies.** Many recent works have explored how LLMs can be used for planning (Yao et al., 2023b; Hao et al., 2023; Zhao et al., 2024; Liu et al., 2023b). A common approach is to provide the LLM with a text-based description of the environment state as input and then query it to produce an action. After executing the action, the resulting text-based state is fed back into the model, creating an interactive loop. Vision-language models have also been applied in a similar manner (Waytowich et al., 2024; Paglieri et al., 2024; Ruoss et al., 2025; Cloos et al., 2024), except that they are prompted with images of the environment state rather than text-based descriptions.

Despite these advances, many frontier LLMs still struggle with spatial reasoning and are prone to hallucinations, which limit their reliability in planning settings. To mitigate these issues, some approaches augment LLM agents with external modules or tools (Cao et al., 2025), fine-tune models on trajectory data (Gaven et al., 2024), incorporate memory modules, or prompting techniques enabling the agent to better structure its reasoning over time. We compared TheoryCoder-2 with agents that use the LLM as the implicit planner (Yao et al., 2023b; Wei et al., 2022; Yao et al., 2023a). We found that while such methods can enhance reasoning, they often suffer from high compute costs, as reasoning models take considerable time to generate answers (Hassid et al., 2025).

**Program Synthesis.** Several works have used program synthesis to build explicit world models of the environment (Tang et al., 2025; Ahmed et al., 2025; Piriyakulkij et al., 2025; Liu et al., 2025), demonstrating improved reasoning capabilities (Gupta & Kembhavi, 2023) compared to standard large language models. EMPA (Tsividis et al., 2021) also uses program synthesis, though it represented the environment in VGDL rather than a general-purpose programming language. Wong et al. (2024) showed that LLMs can be used to learn operators for simple language-instruction domains. Liu et al. (2023a) used LLMs to generate PDDL files and showed that in-context learning examples are important for quality generation. Other work has investigated using vision-language models to learn predicates (Liang et al., 2025), but these methods relied on labeled images to guide object identification, limiting their autonomy. In contrast, our approach targets the end-to-end problem of generating both the goals and the abstractions needed for hierarchical planning (assuming access to a text-based observation of the environment's frame).

**Abstraction Learning in RL.** While our focus is on program-synthesis agents and directly comparable LLM-based agents, abstraction learning and hierarchical planning have also been a long-standing research topic in general reinforcement learning. Key concepts introduced in the options framework (Sutton et al., 1999; Bacon et al., 2017), Feudal RL (Dayan & Hinton, 1992; Vezhnevets et al., 2017), and sub-goal generation (Schmidhuber & Wahnsiedler, 1993; Bakker & Schmidhuber, 2004) remain central in modern deep RL research, including in the offline imitation learning setting (Shiarlis et al., 2018; Kipf et al., 2019; Lu et al., 2021; Gopalakrishnan et al., 2023). Similarly, many recent methods have pushed the sample efficiency of purely neural network model-based RL (Schrittwieser et al., 2020; Hafner et al., 2023), matching human learners' efficiency in certain domains (Ye et al., 2021). However, in general, deep RL methods still remain much less sample efficient, and have lower generalization abilities, compared to neurosymbolic and program synthesis-based agents as have been reported by prior work (Tang et al., 2025; Tsividis et al., 2021). Here, our contribution was to push the current limitation of such a neurosymbolic approach.

## 7 Conclusion

We expanded the scope and efficiency of TBRL by enabling abstraction induction and reuse—a critical step towards making TBRL free of human engineering. We experimentally demonstrated that a novel TBRL system, TheoryCoder-2, is capable of gradually learning reusable abstractions, yielding both improved sample efficiency and solution rates over several baseline LLM agents based on LLMs. Future work will extend TBRL further to make it applicable to environments beyond those with object-oriented, text-based state representations.

## REPRODUCIBILITY STATEMENT

We will clean up the code and release it in a public GitHub repository upon acceptance, including all the prompts used in our experiments (which are also provided in Appendix A). Our codebase builds on the publicly available code of the original TheoryCoder (Ahmed et al., 2025).

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

## A  LANGUAGE MODEL PROMPTS

Here we provide all the prompts used in our experiments, as follows:

- Box 2: Prompt used to Generate PDDL Files
- Box 3: An example of an in-context example for PDDL generation
- Box 4: Another example of an in-context example for PDDL generation

- Box 5: One more example of an in-context example for PDDL generation
- Box 6: Prompt to generate Python predicates
- Box 7: Prompt for abstraction transfer (to generate only the problem file)
- Box 8: Prompt to generate low-level world model
- Domain descriptions for Labythinth and Maze (Box 11), Sokoban (Box 9), BabyAI (Box 10) and Minihack (Box 12 and 13)

---

**Box 2: Generate PDDL Files Prompt**

```
You are an agent playing a 2D grid game, whose raw state is shown
    below.

Can you give me a minimal PDDL domain and problem file for this
    setup that will allow the agent to win the game? Think in terms
    of the most minimal abstract files you can.

Each object in your PDDL problem file is named using ONLY the keys
    of the raw state dictionary.

DO NOT PROPOSE PREDCIATES THAT IMPLY SPATIAL RELATIONS LIKE FROM OR
    TO!!!!!

Domain Description:

{domain_description)

In your PDDL problem file do not represent configuration attributes
    when writing objects
for example for unopened_black_jar you can represent it as black_jar
    . See example 2 and example 3!!!!!!!

Feel free to propose multiple operators, and predicates at once.
    Also you may have two goals in the problem file.

The raw state dictionary keys are considered traversable.

Return your code blocks with
```pddl ```
markup tags so I can easily extract it.

Do NOT use symbols like "-" for the predicate names. For example,
    the predicate "avatar-at" should NOT be proposed since it has a
    "-".
Please use all LOWERCASE for the operator names as well!!!!!!!!!!

Do not name your PDDL DOMAIN FILE DO NOT name predicates:
if, else, in, def, or any other typical python name

predicate names should exactly be one word no underscores in it
    either

Few shot example of what a nice abstraction domain and problem file
    should look like:

{few_shot_PDDL_file_examples}

Raw state of game (generate the files for this):

{raw_state}
```

```
YOUR CURRENT DOMAIN FILE YOU HAVE SYNTHESIZED:

{current_domain}
```

**Box 3: Few Shot PDDL Example 1**

```
state is 'table': [3, 4], 'mug', [4, 4]

(define (domain toy-domain)
  (:requirements :strips :typing)

  (:types
    object
  )

  (:predicates
    (ontop ?x - object ?y - object)
  )

  (:action placeontopof
    :parameters (?obj1 - object ?obj2 - object)
    :precondition (not (ontop ?obj1 ?obj2))
    :effect (ontop ?obj1 ?obj2)
  )
)

(define (problem toy-problem)
  (:domain toy-domain)

  (:objects
    table mug - object
  )

  (:init
    ;; Initially nothing is on top of anything
    (not (ontop mug table))
  )

  (:goal
  ; mug will overlap with table 'table': [4, 4], 'mug', [4, 4]
    (ontop mug table)
  )
)
```

**Box 4: Few Shot PDDL Example 2**

```
state is 'agent': [3, 4], 'apple': [5, 4], 'vines': [4,4], 'axe
   '[1,4], 'unopened_black_jar: [0,1]'

(define (domain toy-domain)
  (:requirements :strips :typing)

  (:types
    object
  )

  (:predicates
    (eaten ?x - object ?y - object)
```

```
  )

  (:action eat
    :parameters (?obj1 - object ?obj2 - object)
    :precondition (not (eaten ?obj1 ?obj2))
    :effect (eaten ?obj1 ?obj2)
  )
)

(define (problem toy-problem)
  (:domain toy-domain)

  (:objects
    agent apple black_jar - object
  )

  (:init
    (not (eaten agent apple))
  )

  (:goal
    (eaten agent apple)
  )
)
```

**Box 5: Few Shot PDDL Example 3**

```
BEGIN EXAMPLE 3

state is 'agent': [6, 3], 'blocked_gold_window': [4,4], '
    unblocked_silver_window': [1,4]

(define (domain toy-domain)
  (:requirements :strips :typing)

  (:types
    object
  )

  (:predicates
    (unblocked ?x - object)
  )

  (:action clear
    :parameters (?x - object)
    :precondition (not (unblocked ?x))
    :effect (unblocked ?x)
  )
)

(define (problem toy-problem)
  (:domain toy-domain)

  (:objects
    agent apple gold_window - object
  )

  (:init
    ; the end goal is to eat the apple
    (not (unblocked gold_window))
```

```
    )

    (:goal
      (unblocked gold_window)
    )
)
```

**Box 6: Python Predicate Generate Prompt**

```
You are a software engineer that must write python predicates. These
    python predicates have to be python versions of the PDDL
    operators that are functions which take the states and arguments
     and returns either True or False. You will need to write Python
     predicates for all the predicates you see in the domain file.
    The problem file, and Raw State is also given to help guide you.

Return your code blocks with
'''python '''
markup tags so I can easily extract it.

BEGIN EXAMPLE

predicate: predicate: isLeftOfop(arg1, arg2)

def isLeftOf(state, arg1, arg2):
    """
    Returns True if arg1 is to the left of arg2, based on their x-
        coordinates.

    Parameters:
    - state: dict with keys as object names and values as [x, y]
        positions
    - arg1: object name (e.g., 'book')
    - arg2: object name (e.g., 'lamp')

    Returns:
    - bool: True if arg1's x-coordinate is less than arg2's, False
        otherwise
    """
    pos1 = state.get(arg1)
    pos2 = state.get(arg2)
    if pos1 is None or pos2 is None:
        return False
    return pos1[0] < pos2[0]

END EXAMPLE

Make sure that you always have state as one of the arguments!

Only synthesize the predicate you see in the domain file and make
    sure
to give it the same name!

Domain File:

{domain_file}

Problem File:

{problem_file}
```

```
Raw State:

{raw_state}

Current Python Low Level World Model:

{world_model}

Game Description:

{game_description}
```

**Box 7: Transfer Abstraction (Generate Only Problem File)**

```
You are an agent playing a 2D grid game, whose raw state is shown
    below.

Can you give me a PDDL problem file for this given PDDL domain file
    that will allow the agent to win the game?
Each object in your PDDL problem file is named using ONLY the keys
    of the raw state dictionary.

You are allowed to specify multiple goals in your problem file!
    Please think about the preconditions and effects carefully.

MAKE SURE TO DOUBLE CHECK AT THE END THAT YOU SPECIFIED MULTIPLE :
    GOALS IN THE PROBLEM FILE
JUST BECAUSE YOU HAVE ONE MISSION DOESN'T MEAN YOU WILL USE THAT
    MISSION AS THE SINGLE GOAL

for example you would maybe need to do a subgoal in order to achieve
     the mission!! DO NOT JUST ASSUME ONE GOAL in problem file

Return your code blocks with
'''pddl '''
markup tags so I can easily extract it.

{few_shot_PDDL_file_examples}

Domain file you need to use (generate the problem file for this):

{domain_file}

Raw state of game (generate the problem file for this):

{raw_state}

MISSION:

{mission}

Domain Description:

{domain_description}
```

**Box 8: Generate Low level World Model**

```
You are an AI agent that must come up with a transition model of the
    game you are playing.

A BFS low-level planner that will use your synthesized transition
    model to find the low-level actions that will allow you to win
    levels of the game.

You are also given state transition after executing random actions
    that will help as well.
Note that if there is no change returned after doing that action, it
    means that moving was prevented somehow such as by an obstacle.

DESCRIPTION OF DOMAIN:

{domain_description}

CURRENT STATE:

{current_state}

ACTION SPACE:

{actions_set}

Replay Buffer (last {num_random_actions} transitions):

{errors_from_world_model}

UTILS:

{utils}

RESPONSE FORMAT:

- Make sure you use .get() to access the dictionary to avoid key
    errors!
For example:
avatar_pos = new_state.get('avatar') to get avatar pos
cake_pos = new_state.get('cake') to get cake pos

```python

# make sure to include these import statements
from utils import directions

def transition_model(state, action):

    Return State
```
```

### Box 9: Sokoban Domain Description

```
In this domain, you have to push the boxes into the holes to win. If
    you push the box into the hole, the box will disappear.
```

---

### Box 10: BabyAI Domain Description

```
The agent needs to navigate the maze to win. If the agent is facing
    a key, it can pick it up.
The agent can also unlock doors in which case the door will become
open_COLORNAME_door in the state.

For this environment the state key `agent_carrying` is a list of
    object names the agent currently holds (e.g., `['red_key']`).
When a door is unlocked it will turn from locked_ to open_ (e.g., '
    locked_blue_door' -> 'open_blue_door').
When a closed door is opened it will turn from closed_ to open_ (e.g
    ., 'closed_blue_door' -> 'open_blue_door').

You can toggle any closed doors to open them and locked ones when
    you have their COLOR_key

You cannot move forward through closed_ doors unless they are _open
So you will need to toggle them
so closed_ doors are essentially similar to grey walls in that they
    block you

In the game you cannot overlap with any objects, to pickup the key
    you need to be adjacent to it and facing it.
```

---

### Box 11: Maze and Labyrinth Domain Description

```
In this domain, you control the avatar and need to reach the goal.
If you touch a trap you will die.
```

---

### Box 12: Minihack Navigation Levels Description

```
In this domain, move the agent to the downstair case.
```

---

### Box 13: Minihack Wand Of Death Description

```
Kill the minotaur to win.
To use the wand you zap, then select_f and finally shoot in the
    direction you want.
You need to be about 5 squares away from the minotaur to kill it
    with the zap.
It's good to sequence your actions by zapping, selecting f and then
    pointing to the correct direction.
In order to shoot_ in a particular direction, you must always first
    zap, then select_f and then shoot in that direction – all as one
     action sequence.
```

## B  RESULTS AND OUTPUTS

Here we provide some of the results and outputs for TheoryCoder-2's learned abstractions and low-level world model. Additionally, we provide the hand-designed oracle abstractions. Notably, the the learned and hand-designed abstractions are similar in their preconditions and effects.

- Box 14: human hand-designed abstractions for BabyAI
- Box 15: TheoryCoder-2's learned abstractions for BabyAI

- Box 16: TheoryCoder-2's learned Low_level world model for BabyAI

### Box 14: Human Hand-Designed Abstractions for BabyAI

```
(:action pickup
    :parameters (?key – object)
    :precondition (not (holding ?key))
    :effect (holding ?key)
  )

  (:action unlock
    :parameters (?door – object ?key – object)
    :precondition (and (holding ?key) (not (unlocked ?door))
    (unlocks ?key ?door))
    :effect (unlocked ?door)
  )
```

### Box 15: TheoryCoder-2's Learned Abstractions for BabyAI

```
(:action pickup
    :parameters (?agent – object ?item – object)
    :precondition (not (holding ?agent ?item))
    :effect (holding ?agent ?item)
  )

(:action unlock
    :parameters (?agent – object ?door – object ?key – object)
    :precondition (and (holding ?agent ?key) (not (unlocked ?door)))
    :effect (unlocked ?door)
  )
```

### Box 16: TheoryCoder-2's Learned Low-level world model for BabyAI

```python
from utils import directions

def transition_model(state, action):
    # Copy the current state to avoid modifying the original state
    new_state = state.copy()

    # Get agent position, direction, and carrying status
    agent_pos = new_state.get('red_agent')[0]
    agent_direction = new_state.get('agent_direction')
    agent_carrying = new_state.get('agent_carrying', [])

    # Define movement based on current direction
    def move_forward(pos, direction):
        return [pos[0] + direction[0], pos[1] + direction[1]]

    # Action: 'left'
    if action == 'left':
        # Rotate direction counter-clockwise
        if agent_direction == [0, 1]:  # Facing up
            new_state['agent_direction'] = [-1, 0]  # Facing left
        elif agent_direction == [-1, 0]:  # Facing left
            new_state['agent_direction'] = [0, -1]  # Facing down
        elif agent_direction == [0, -1]:  # Facing down
            new_state['agent_direction'] = [1, 0]  # Facing right
        elif agent_direction == [1, 0]:  # Facing right
            new_state['agent_direction'] = [0, 1]  # Facing up
```

```
        # Action: 'right'
        elif action == 'right':
            # Rotate direction clockwise
            if agent_direction == [0, 1]:  # Facing up
                new_state['agent_direction'] = [1, 0]  # Facing right
            elif agent_direction == [1, 0]:  # Facing right
                new_state['agent_direction'] = [0, -1]  # Facing down
            elif agent_direction == [0, -1]:  # Facing down
                new_state['agent_direction'] = [-1, 0]  # Facing left
            elif agent_direction == [-1, 0]:  # Facing left
                new_state['agent_direction'] = [0, 1]  # Facing up

        # Action: 'forward'
        elif action == 'forward':
            new_pos = move_forward(agent_pos, agent_direction)
            if new_pos not in new_state.get('grey_wall', []):
                for door_type in ['locked_', 'closed_']:
                    for key in new_state.keys():
                        if key.startswith(door_type) and new_pos in
                            new_state[key]:
                            return new_state  # Movement blocked by a
                                closed or locked door
                new_state['red_agent'] = [new_pos]

        # Action: 'pickup'
        elif action == 'pickup':
            target_pos = move_forward(agent_pos, agent_direction)
            for key, positions in new_state.items():
                if 'key' in key and target_pos in positions:
                    new_state['agent_carrying'].append(key)
                    positions.remove(target_pos)
                    break

        # Action: 'toggle'
        elif action == 'toggle':
            target_pos = move_forward(agent_pos, agent_direction)
            for door_type in ['closed_', 'locked_']:
                for key, positions in new_state.items():
                    if key.startswith(door_type) and target_pos in
                        positions:
                        color = key.split('_')[1]
                        if door_type == 'closed_' or f'{color}_key' in
                            agent_carrying:
                            new_state[f'open_{color}_door'] = positions
                            del new_state[key]
                        break

        # Action: 'drop'
        elif action == 'drop':
            if agent_carrying:
                dropped_key = agent_carrying.pop()
                new_state['agent_carrying'] = agent_carrying
                new_state[dropped_key] = [move_forward(agent_pos,
                    agent_direction)]

        return new_state
```