# OpenReview forum: "Learning Abstractions for Hierarchical Planning in Program-Synthesis Agents"
_ICLR.cc/2026/Conference — Submitted to ICLR 2026_

### Official Review · Reviewer_LESQ · 2025-10-24

**Soundness:** 3
**Presentation:** 3
**Contribution:** 3
**Rating:** 6
**Confidence:** 3

**Summary:**

TheoryCoder-2 is a theory-based reinforcement learning (TBRL) agent that leverages LLMs’ in-context learning to automatically synthesize reusable high-level abstractions from experience and integrate them into hierarchical planning for program-synthesis agents. It is evaluated on environments including BabyAI and VGDL games (e.g., Sokoban), compared to LLM-augmented classical planning, reasoning-based planners, and prior program-synthesis agents. TheoryCoder-2 demonstrates substantially improved sample efficiency and generalization—solving complex tasks that baselines fail—while requiring minimal human guidance.

**Strengths:**

- Innovative integration of LLMs and TBRL. TheoryCoder-2 advances theory-based RL by using LLMs to learn high-level abstractions (PDDL operators) rather than relying on hand-coded ones.

- Hierarchical coupling of learned abstractions + executable world model. The combination of PDDL-level planning with a learned Python transition model (plus predicate classifiers) is a principled way to ground abstractions for execution.

- Comprehensive evaluation with multiple baselines, ablation studies, and diverse metrics (token cost, compute time, success rate)

**Weaknesses:**

- Limited analysis of abstraction quality. While the paper highlights the ability to actively learn reusable abstractions rather than relying on human-provided ones, it remains unclear what kinds of abstractions are newly discovered beyond those already captured by hand-specified ones. The paper would benefit from a deeper analysis showing whether the learned abstractions differ qualitatively or functionally from human-designed ones, and to what extent they are more general, compositional, or transferable.

- Limited domain diversity / ecological validity. Experiments are restricted to VGDL grid games and BabyAI — discrete, object-oriented toy domains.

**Questions:**

1. Is it possible to incorporate objective measures of abstraction quality—such as correctness, granularity, or redundancy—to more rigorously assess the learned abstractions?

2. Could the paper include concrete examples or comparisons showing cases where the learned abstractions outperform or complement human-specified ones?

3. The solution rate of TheoryCoder-2 appears to remain below 90%, could the paper analyze or discuss the primary failure reasons?

4. Is it feasible to adapt or extend TheoryCoder-2 to more diverse or ecologically valid domains?

---

> ### Author Response · Authors · 2025-12-03
> **Response to Reviewer LESQ**
>
> We thank the reviewer for their valuable feedback and suggestions which help us improve our work. We address the remaining concerns as follows:
>
> > (Weakness) Limited analysis of abstraction quality. While the paper highlights the ability to actively learn reusable abstractions rather than relying on human-provided ones, it remains unclear what kinds of abstractions are newly discovered beyond those already captured by hand-specified ones. The paper would benefit from a deeper analysis showing whether the learned abstractions differ qualitatively or functionally from human-designed ones, and to what extent they are more general, compositional, or transferable.
>
> The reviewer raises an interesting point. We have extended our analysis to include human-designed abstractions and included it as an Oracle solution. We also now provide an analysis discussing the qualitative differences between the abstractions that are learned by TC2 and the human-designed ones. We find that they are of comparable quality. However, the learned abstractions sometimes offered clever reusability that we did not consider. For example, we had hand-designed a push_to abstraction for Sokoban and a move_to abstraction for Labyrinth and Maze, whereas TC2 learned to reuse moveontop from these domains for Sokoban.
>
> > Limited domain diversity / ecological validity. Experiments are restricted to VGDL grid games and BabyAI — discrete, object-oriented toy domains.
>
> We have now extended our experiments to include Minihack—it is still a grid-world but contains popular and challenging environments. Future work would need to be done to extend TC2 to handle continuous non-grid domains. We provide a discussion of this limitation (Sec 5) in our paper.
>
> > (Q1) Is it possible to incorporate objective measures of abstraction quality—such as correctness, granularity, or redundancy—to more rigorously assess the learned abstractions?
>
> This is an interesting and challenging question, and we have added some analysis in this direction. We qualitatively compare and analyze the abstractions in Section 5 and Appendix B. However, in general, systematic evaluation of learned abstractions is difficult beyond the evaluation in the actual environments. For example, it is an open question on how to properly assess semantics of a particular abstraction.
>
> > (Q2) Could the paper include concrete examples or comparisons showing cases where the learned abstractions outperform or complement human-specified ones?
>
> Thank you for pointing this out. We have added the Oracle hand-designed abstractions in our analysis. We find that abstractions learned by TC2 are of comparable quality. However, the learned abstractions sometimes offered clever reusability that we did not consider. For example, we had hand-designed a push_to abstraction for Sokoban and a move_to abstraction for Labyrinth and Maze, whereas TC2 learned to reuse moveontop from these domains for Sokoban.
>
> > (Q3) The solution rate of TheoryCoder-2 appears to remain below 90%, could the paper analyze or discuss the primary failure reasons?
>
> We include a more thorough analysis of this in the discussion description. The errors we observed are only with the low-level world model program (which is not specific to TC2). We speculate that further program revision iterations would likely improve accuracy; however, we did not investigate this possibility in this work.
>
> > (Q4) Is it feasible to adapt or extend TheoryCoder-2 to more diverse or ecologically valid domains?
>
> We now include results on Minihack, which is a more challenging RL benchmark. In future work, we also plan to extend our system with a visual front-end that translates high-dimensional state-spaces (e.g. pixels) to more abstract, object-oriented representations like those in our current eval set, but this is out of scope of this work.
>
> We believe our response resolves all the main concerns. Thank you very much.

---

### Official Review · Reviewer_CgvW · 2025-11-01

**Soundness:** 3
**Presentation:** 2
**Contribution:** 2
**Rating:** 4
**Confidence:** 4

**Summary:**

This paper presents TheoryCoder-2, a new TBRL method that learns reusable abstractions for hierarchical planning. Unlike prior systems such as TheoryCoder (Ahmed et al., 2025), which rely on manually provided PDDL abstractions, TheoryCoder-2 uses LLMs to synthesize abstractions from experience via in-context learning. The basic idea is to incrementally grow a library of abstractions. Empirically, TheoryCoder-2 has better sample efficiency and generalization compared to baseline LLM-based planners (LLM + P, WorldCoder, reasoning LLMs), solving tasks that others fail on, while requiring minimal human input.

**Strengths:**

-  The paper addresses a limitation in TBRL and many program-synthesis agents (dependency on human-input). The proposed approach is an interesting solution to this issue.

- To the best of my knowledge, the combination of (i) PDDL abstraction synthesis via in-context LLM prompting, (ii) creation of a curriculum-based abstraction collection, and (iii) grounding through low-level Python models is novel.

- The idea fits ICLR.

**Weaknesses:**

To me, the main weakness of the paper is the scope of the experimental analysis. The experiments are not sufficient for a strong, general conclusion that TheoryCoder-2 achieves scalable, human-like abstraction learning. There is only one curriculum with eight problems evaluated in the experimental results, so I am not confident taking conclusions from it.

Usually, I think that good ideas can overcome bad experimental results, but in the case of this paper, the main motivation is to show that the method "is a significant improvement in the applicability of TBRL, and an important step towards building AI systems that learn like humans". For that, I would need to see much more challenging and meaningful benchmarks.

On top of that, I am quite confused by section 4.1 First, comparing Sokoban to Maze and Labyrinth is like comparing the Earth with a soccer ball. I find it quite hard to believe that the `moveontop` abstraction would not *hurt* performance in Sokoban.The fact that Sokoban seems much easier (from the raw data) than the other two, makes me think that the task used is not appropriately challenging. Additionally, Labyrinth and Maze are just isomorphic problems, so I find the presence of both in the curriculum a bit underwhelming.

Last, a minor complain about the presentation: I think the paper deserves a detailed revision and editing. In the first half of the paper, in particular, I found that many passages had words that were used with some very ad-hoc meaning. While this is often harmless and just a matter of style, it gave me an impression that the paper had been rushed and the text was too vague for a scientific paper.

Example: (italics are mine)

> The domain file specifies abstract actions (called “operators”, e.g., “open door”) and their preconditions/effects, as well as abstract
states (e.g., “door unlocked”) which are *summarized* through Boolean predicates that capture task-relevant features. The problem file specifies the initial state and goal conditions for a particular task. Together, these files are *consumed* by a *classic* PDDL planner.

None of the marked words is intended to mean what it says in the text: "summarized" means "fully described"; "consumed" simply means "read'; "classic" is probably just a typo and I assume you mean a "classical planner". (And, of course, the description of a PDDL domain and instances is incomplete.)

**Questions:**

Could you please explain me the Sokoban task you had in the curriculum and how the moveontop abstraction actually help there? I am clearly missing something.

Do you have results for other curricula that do not involve moving on a grid?

---

> ### Author Response · Authors · 2025-12-03
> **Response to Reviewer CgvW**
>
> We thank the reviewer for their valuable feedback and suggestions which help us improve our work. We address the remaining concerns as follows:
>
> > To me, the main weakness of the paper is the scope of the experimental analysis. The experiments are not sufficient for a strong, general conclusion that TheoryCoder-2 achieves scalable, human-like abstraction learning. There is only one curriculum with eight problems evaluated in the experimental results, so I am not confident taking conclusions from it. Usually, I think that good ideas can overcome bad experimental results, but in the case of this paper, the main motivation is to show that the method "is a significant improvement in the applicability of TBRL, and an important step towards building AI systems that learn like humans". For that, I would need to see much more challenging and meaningful benchmarks.
>
> Thank you for pointing this out. To address this concern, we have added another curriculum for the popular Minihack domain [1] to demonstrate the broader applicability of TBRL—we hope this assuages the reviewer’s main concerns. In addition, we should emphasize that we do not consider this work to have fully solved the problem of recreating human-like abstraction learning! Rather, the central goal of this work is to build an extension to prior work that removes the human from the loop in a crucial way.
>
> [1] MiniHack the Planet: A Sandbox for Open-Ended Reinforcement Learning Research.
> Mikayel Samvelyan, et al. NeurIPS 2021 Datasets and Benchmarks Track.
>
> > On top of that, I am quite confused by section 4.1 First, comparing Sokoban to Maze and Labyrinth is like comparing the Earth with a soccer ball. I find it quite hard to believe that the moveontop abstraction would not hurt performance in Sokoban.The fact that Sokoban seems much easier (from the raw data) than the other two, makes me think that the task used is not appropriately challenging. Additionally, Labyrinth and Maze are just isomorphic problems, so I find the presence of both in the curriculum a bit underwhelming.
>
> We would like to point out that the purpose of this experiment was to specifically test how the abstraction can be reused for isomorphic problems. Additionally, to address the reviewer’s concern, we have included Minihack experiments that learn the move_to abstraction on a 5x5 room and reuses it for a 15x15 room and variations where there are monsters and traps.
>
>
> > Last, a minor complain about the presentation: I think the paper deserves a detailed revision and editing. In the first half of the paper, in particular, I found that many passages had words that were used with some very ad-hoc meaning.
>
> We agree that our language should be precise. We have attempted to improve the writing in this way. We have specifically changed the wordings from “summarized” to “represented” AND from “consumed” to “taken as input by a classical planner”. We would also like to point out that other reviewers found that the text is well written, so we hope these minor edits are sufficient to improve the clarity.
>
> > (Question) Could you please explain me the Sokoban task you had in the curriculum and how the moveontop abstraction actually help there? I am clearly missing something.
>
> We apologize if the reviewer found our Sokoban task description confusing. To clarify, we purposefully chose a very simplified version of Sokoban with one box and one hole. This task was chosen specifically to see how abstractions learned from a different task, such as navigating a maze, can be used for a task that requires pushing a box into a hole. The central aim of this task is to validate that moveontop is reusable in other settings, rather than to show performance improvements for Sokoban.
>
> > (Question) Do you have results for other curricula that do not involve moving on a grid?
>
> Not yet. This will require certain non-trivial extensions, which we aim to build in future work. Please see also our discussion above under R2’s comments for existing work that establishes promising starting points.
>
> We believe our response resolves all the main concerns. Thank you very much.

---

### Official Review · Reviewer_u2HK · 2025-11-01

**Soundness:** 3
**Presentation:** 4
**Contribution:** 2
**Rating:** 4
**Confidence:** 3

**Summary:**

The paper introduces TheoryCoder-2, a system that solves agent tasks through learning reusable abstractions to solve new tasks. It focuses on the domain of the TBRL paradigm, which attempts to model the environment and interactions through program synthesis. This work builds on theoryCoder, which represents theory through Python programs. It mainly improves in the way that it does not require hand-crafted abstractions. Instead, it only needs a task-agnostic, high-level prompt and enables the system to automatically learn abstractions through in-context learning ability of language models. The system is also able to grow its learned library through further interaction with the environment. The paper evaluates on environments such as BabyAI and VGDL games, and shows that their method has good accuracy with  great sampling efficiency, compared to other frameworks that also utilize LLMs.

**Strengths:**

- The paper is well-written and very easy to follow. It introduces a domain, the task, and method in a way that is friendly to the broader RL/ML community.
- The improvement from TheoryCoder to TheoryCoder2 is significant. It gets rid of domain specific human annotation, which shows potential to scale to more complex, unseen environments
- Using programs as a representation of complex environments and their transitions is interesting and shows a direction of world model.

**Weaknesses:**

- Though the modeling through programs works well on these environments, I am unsure about how the TBRL paradigm and TheoryCoder2 will perform on more complex, real-world tasks. Recent advancements on Agent Task have proposed more complex tasks e.g., ALFWorld, Webshop, WebArena. I doubt that this paradigm and method can still discover and model the transition function. This is because purely modeling the very complex environment with code is difficult (if not impossible). The authors may want to show that such method may also work in more complex environments through experiments.

- The experiment is not sufficient. More specifically, it chooses weaker baseline methods for comparison. For example, (according to citation), ReAct is used as a baseline to be compared. While ReAct is a famous paradigm, several methods have been proposed based on it, and have achieved better results. The authors may need to add experiments that compare more stronger baseline. Moreover, it seems that the LLM used for different methods are different: some baseline methods use o4-mini and the proposed method use GPT-4o, the author may consider keep them consistent.

**Questions:**

- See weaknesses.

- Could the authors elaborate on the potential to extend this framework to embodied tasks?

---

> ### Author Response · Authors · 2025-12-03
> **Response to Reviewer u2HK**
>
> We thank the reviewer for their valuable feedback and suggestions which help us improve our work. We address the remaining concerns as follows:
>
> > (Weakness) Though the modeling through programs works well on these environments, I am unsure about how the TBRL paradigm and TheoryCoder2 will perform on more complex, real-world tasks. Recent advancements on Agent Task have proposed more complex tasks e.g., ALFWorld, Webshop, WebArena. I doubt that this paradigm and method can still discover and model the transition function. This is because purely modeling the very complex environment with code is difficult (if not impossible). The authors may want to show that such method may also work in more complex environments through experiments.
>
> Thank you for this question regarding the level of complexity that TBRL can handle. To partially address this concern, we have now added MiniHack [1] environments, and find that TC2 also performs well.  Thank you also for suggesting specific environments; however, they are not compatible with the main goal of this work as they are focused on different classes of task, or in the case of ALFWorld, the abstractions are already prespecified and reported in the observation state.
>
> [1] MiniHack the Planet: A Sandbox for Open-Ended Reinforcement Learning Research.
> Mikayel Samvelyan, et al. NeurIPS 2021 Datasets and Benchmarks Track.
>
>
> > The experiment is not sufficient. More specifically, it chooses weaker baseline methods for comparison. For example, (according to citation), ReAct is used as a baseline to be compared. While ReAct is a famous paradigm, several methods have been proposed based on it, and have achieved better results. The authors may need to add experiments that compare more stronger baseline.
>
> Unfortunately, we do not fully understand this criticism, and we respectfully disagree that stronger baselines are missing. Our baselines do include a comprehensive set of representative and state-of-the-art baselines such as WorldCoder, not simply ReAct. Did the reviewer have any specific baseline in mind other than ReAct?
>
> > Moreover, it seems that the LLM used for different methods are different: some baseline methods use o4-mini and the proposed method use GPT-4o, the author may consider keep them consistent.
>
> Thank you for pointing this out. We added GPT-4o as a non-reasoning model baseline (see Figure 3). We would like to point out that the original reason some methods use GPT-4o is because of the intention to test whether a non-reasoning model can be used to quickly generate the world model components.
>
> > (Question) Could the authors elaborate on the potential to extend this framework to embodied tasks?
>
> An existing body of work [1, 2] develops techniques for synthesizing PDDL operators and predicates for embodied robotics planning tasks, including robotic arm manipulation. In principle, TC2 can be extended to embodied tasks by building on these techniques.
>
> [1] Liu, W., Nie, N., Zhang, R., Mao, J., & Wu, J. Learning compositional behaviors from demonstration and language. Conference on Robot Learning. 2024
>
> [2 ]Li, A., & Silver, T. Embodied active learning of relational state abstractions for bilevel planning. Preprint arXiv:2303.04912. 2023
>
> We believe our response resolves all the main concerns. Thank you very much.

---

### Official Review · Reviewer_QELz · 2025-11-02

**Soundness:** 2
**Presentation:** 3
**Contribution:** 2
**Rating:** 4
**Confidence:** 3

**Summary:**

This paper introduces TheoryCoder-2, a theory-based reinforcement learning (TBRL) agent that learns reusable abstractions from experience via large language models’ in-context learning ability. Unlike previous systems that rely on hand-crafted abstractions, TheoryCoder-2 autonomously synthesizes and grounds abstractions for hierarchical planning. Experiments on BabyAI and VGDL games demonstrate performance improvement in sample efficiency and generalization over LLM-based and program-synthesis baselines.

**Strengths:**

- The paper focuses on an interesting and timely direction: automatically learning reusable abstractions with LLMs.
- The presentation is clear and the paper is generally easy to follow.

**Weaknesses:**

- The method resembles prior work on learning and reusing skills through curriculum learning and code-based representations. The main difference here is expressing these skills in the PDDL format, but similar high-level ideas (e.g., learning, storing, and recalling code snippets or functions for reuse), have been explored in many recent works such as [1]. However, these related baselines are not discussed in sufficient depth, which makes it hard to assess what is genuinely new beyond the specific implementation.
- The paper does not include a direct comparison between TheoryCoder-2 and human-coded abstractions (e.g., hand-crafted PDDL operators). Without this comparison, it is difficult to understand the gap between the generated abstractions vs human-written ones.

---
[1] Liu, Anthony Z., et al. "Interactive and Expressive Code-Augmented Planning with Large Language Models." arXiv preprint arXiv:2411.13826 (2024).

**Questions:**

Have the author tried running with different model scales (e.g., GPT-4o vs smaller models). How sensitive is the abstraction quality to model size or reasoning capacity?

---

> ### Author Response · Authors · 2025-12-03
> **Response to Reviewer QELz**
>
> We thank the reviewer for their valuable feedback and suggestions which help us improve our work. We address the remaining concerns as follows:
>
> > (Weakness) The method resembles prior work on learning and reusing skills through curriculum learning and code-based representations. The main difference here is expressing these skills in the PDDL format, but similar high-level ideas (e.g., learning, storing, and recalling code snippets or functions for reuse), have been explored in many recent works such as [1]. However, these related baselines are not discussed in sufficient depth, which makes it hard to assess what is genuinely new beyond the specific implementation.
>
> Thank you for pointing this out. Yes, our work has similarities to other recent work, such as REPL-Plan [1], belonging to the family of LLM-based “program synthesis agents”. While not all the baselines are included in our experiments, we do include all the **representative baselines** (see, Figure 1 for a comparison overview), such as LLM + P and WorldCoder. For example, our LLM + P baseline is quite similar to REPL-Plan in that both works synthesize functions for reuse; and our text (Sec 6, paragraph Program Synthesis) already addresses the ways in which our work differs from that. We have added an additional citation to REPL-Plan in the related work section.
>
> > (Weakness) The paper does not include a direct comparison between TheoryCoder-2 and human-coded abstractions (e.g., hand-crafted PDDL operators). Without this comparison, it is difficult to understand the gap between the generated abstractions vs human-written ones.
>
> Following this suggestion, we have rerun the human-coded abstractions and added it to Figure 3, named as “Oracle”. We find that TC2’s learned abstractions are able to achieve high accuracy almost matching the Oracle solution. We agree with the reviewer that this is a useful baseline. Thank you for pointing this out.
>
> > (Question) Have the author tried running with different model scales (e.g., GPT-4o vs smaller models). How sensitive is the abstraction quality to model size or reasoning capacity?
>
> We previously tried open-source models and found that they perform much worse in terms of both reasoning capacity and abstraction generation than closed-source models. For this reason, we chose to focus on closed-source models. Future work could more systematically investigate model size on abstraction quality, but it is out of scope for the current work: our main contribution here is to show that such abstraction learning is feasible.
>
> We believe our response resolves all the main concerns. Thank you very much.

---

### Author Response · Authors · 2025-12-03
**General Response to Reviewers**

We thank all the reviewers for their valuable time reviewing our work, and suggestions which helped us substantially improve our work.

We updated the manuscript with the following revisions (in PDF, the main changes in text are highlighted in color/violet).

We have added new experimental results on Minihack:
- Figure 2 updated to include new Minihack levels
- Accordingly updated Sec 4.1
- Accordingly updated Table 1
- Accordingly updated Figure 3

Other changes are:
- Algorithm box 1 describing our method
- Updated Figure 3 with the GPT-4o baseline (LLM non-reasoning)
- Updated Figure 3 with the Oracle (hand-designed abstractions) baseline
- Additional citations as suggested by the reviewers
- Discussion on quantifiable abstraction quality metrics in the discussion section
- Appendix B added: provides some example results and outputs of learned abstractions and low-level world model, and includes examples of hand-designed abstractions

---

### Meta-Review · Area_Chair_JfTS · 2025-12-28

**Summary:**

This paper introduces TheoryCoder-2, a theory-based reinforcement learning (TBRL) framework that leverages large language models to autonomously synthesize reusable abstractions for hierarchical planning.

Major Concerns

- Limited Novelty Beyond Existing Program-Synthesis Agents (QELz, u2HK). While the authors clarify relationships to prior work (e.g., REPL-Plan, WorldCoder), the core contribution still appears incremental. This concern is reflected in multiple reviewers rating the contribution as only “fair.”

- Insufficient Evidence for Scalability and Generality (u2HK, CgvW, LESQ). The main claim—that TheoryCoder-2 is a significant step toward scalable, human-like abstraction learning—remains only weakly supported. Despite the addition of MiniHack, the experimental evaluation is still largely confined to grid-based, object-centric toy domains. The paper does not convincingly demonstrate that the proposed paradigm can scale to substantially more complex environments.

- Experimental Scope Still Not Commensurate with the Claims
Although the rebuttal improves the experimental coverage, the overall scope remains limited relative to the ambition of the paper. The number of curricula is small, many tasks are closely related or isomorphic, and the evaluation does not yet justify broad claims about learning abstractions “like humans.” Some concerns about baseline strength and fairness are partially addressed, but doubts remain about whether the empirical comparisons are sufficiently comprehensive.

After considering the reviews and the authors’ rebuttal, I recommend rejection.

**Reviewer Concerns:**

The concerns mentioned above are not fully addressed by the rebuttal.

The concern fully addressed is "No direct comparison to human-designed PDDL."

**Reviewer Scores:**

I do not think reviewers QELz, u2HK, CgvW would change the score.

---

### Decision · Program_Chairs · 2026-01-26

Reject